# Ultrasound-Guided Dynamic Needle-Tip Positioning Method Is Superior to Conventional Palpation and Ultrasound Method in Arterial Catheterization

**DOI:** 10.3390/jcm11216539

**Published:** 2022-11-03

**Authors:** Guannan Wu, Chen Chen, Xiaoling Gu, Yanwen Yao, Dongmei Yuan, Jiawen Lv, Beilei Zhao, Qin Wang

**Affiliations:** 1Department of Respiratory Medicine, Jinling Hospital, Nanjing University School of Medicine, 305 East Zhongshan Road, Nanjing 210002, China; 2Department of Respiratory Medicine and Critical Medicine, Jinling Hospital, Nanjing Medical University, 305 East Zhongshan Road, Nanjing 210002, China

**Keywords:** ultrasound, dynamic needle-tip positioning, arterial catheterization, palpation

## Abstract

Background: Dynamic needle-tip positioning (DNTP) was shown to improve arterial cannulation efficiency with fewer complications than conventional palpation and ultrasound methods by some studies. However, this is still controversial, and we performed this meta-analysis to comprehensively assess its value in arterial cannulation. Methods: A literature search of randomized controlled trials was conducted, and 11 studies were finally included. Efficiency outcomes (first-attempt success, overall success, and total cannulation time) and complications (hematoma, thrombosis, posterior wall puncture, and vasospasm) were separately analyzed. Subgroup analyses in different populations under cannulation were also performed. Results: DNTP was associated with increased first-attempt success (pooled RR = 1.792, *p* < 0.001), overall success (pooled RR = 1.368, *p* = 0.001), and decreased cannulation time (pooled SMD = −1.758, *p* = 0.001) than palpation. DNTP gained even more advantage in small children and infants. No significant difference in these outcomes between DNTP and conventional ultrasound method was detected. Fewer hematoma occurred in DNTP than palpation (pooled RR = 0.265, *p* < 0.001) or traditional ultrasound (pooled RR = 0.348, *p* < 0.001). DNPT was also associated with fewer posterior wall punctures (pooled RR = 0.495, *p* = 0.001) and vasospasm (pooled RR = 0.267, *p* = 0.007) than traditional ultrasound. Conclusions: DNTP was a better choice in artery cannulation than conventional palpation and ultrasound method, especially in small children and infants.

## 1. Introduction

Arterial cannulation is commonly used in patients who are critically ill or patients undergoing surgery for clinical monitoring and special treatment, including intra-aortic balloon pump (IABP) and veno-arterial extracorporeal membrane oxygenation (VA-ECMO) [1]. In arterial cannulation practice, improving first-attempt success rate is one of the main goals and the most effective method to reduce complications [2]. Previous studies have indicated that ultrasound-guided arterial catheterization was superior to conventional palpation with a remarkably improved first-attempt success rate [3]. Two different techniques are mainly applied in clinical practice: the long axis in-plane (LA-IP) approach and the short axis out-of-plane (SA-OOP) approach [4]. In recent years, a modified version of the SA-OOP approach termed ‘dynamic needle-tip positioning (DNTP)’ was shown to be a better choice by integrating the advantage of both the SA-OOP and LA-IP approach [5]. By frequently moving an ultrasound probe along the target blood vessel, the DNTP approach could provide an image of both the whole puncture route and surrounding tissue. In 2021, a meta-analysis, including six randomized controlled trials (RCTs) and one retrospective study, indicated that the DNTP method was better in terms of first-attempt success, overall success, and complications than conventional palpation in arteriovenous puncture [6]. However, this study farraginously included five studies of arterial catheterization and two studies of venous catheterization, and no further analysis in arterial catheterization catalog was performed because of the limited number of included studies. Additionally, a few more studies that focused on the DNTP technique and arterial catheterization have been published since then, and we performed this present meta-analysis to evaluate the efficiency of the DNTP approach in arterial catheterization.

## 2. Materials and Methods

### 2.1. Searching Strategy

A literature search of online databases (PubMed, ISI Web of Knowledge, Cochrane Library, MedLine-EBSCO, Elsevier ScienceDirect, pringerLink, and Wiley Online Library) up to 3 September 2022 was conducted by using “ultrasound”, “catheter”, “catheterization”, “cannulation”, “artery”, and “dynamic needle-tip positioning” for full text. Manual searches of citations from these original studies were also carried out. Then, we eliminated unrelated publications and checked these remaining articles to avoid replicated data in different publications from the same author or researching group. Since this current study is not involved in ethics issues, no ethics statement is needed.

### 2.2. Eligibility Criteria

To be included, studies have to meet the following criteria: (1) randomized controlled trial, (2) containing retrievable quantitative data of efficiency and/or complication outcomes, and (3) full text reported in English. Any study that meets either of the following criteria was excluded: (1) prospective observational studies or retrospective studies, (2) arterial cannulation on practice models, such as gelatine phantom, and (3) RCTs assessing the use of Doppler ultrasonography. No limitation of patients’ ages and major arterial catheterization operators was set in this study. Two authors (Wu G and Chen C) independently performed the study screen and assessed the methodological quality of included studies. Risk of publication bias for the individual study is statistically analyzed.

### 2.3. Data Extraction and Quality Assessment

Publications were checked independently by two of the authors (Wu G. and Chen C.). The two authors also independently evaluated and extracted information of eligible publications for the final analysis. Disagreements were resolved by discussion between Wu G. and Chen C. If no consensus could be reached, a third author (Wang Q.) was consulted to reach a conclusive decision. The following data were extracted or calculated: name of first author, year of publication, total numbers of cases and controls, country or continent of subjects in each study, randomization method, patient age range, the male subject percentage, weight, height, body mass index (BMI), name of catheterized artery, diameter and depth of the catheterized artery, mean systolic blood pressure (SBP), mean diastolic blood pressure (DBP), efficiency outcomes (including first-attempt success rate, total success rate, performing time, and the total number of attempts), and cannulation associated complications (including hematoma, thrombosis, spasm, and ischemia). The quality of each included study was reviewed by an established standard from the Cochrane by Revman 5.4.1 (Cochrane, London, UK).

### 2.4. Statistical Synthesis and Analysis

All statistical analysis was performed by using STATA12.0 software (StataCorp, College Station, TX, USA). For binary outcomes, the absolute number of positive and negative ending in case and control groups was extracted, pooled, and analyzed. For continuous variable, the total number of participants, mean, and standard deviation (SD) values were obtained for the case and control groups and analyzed for pooled results. When these values were reported as median and interquartile range or total range, the mean and SD values were estimated by Revman 5.4.1 in accordance with the Cochrane Handbook. Random-effects model was applied in all pooled analysis. Meta-regression analysis was performed in catheter size, mean patient age, mean diameter of the catheterized artery, mean depth of the catheterized artery, mean SBP, and mean DBP to screen possible influencing factors. Publication bias was assessed by Begg’s test and Egger’s test. A value of “Pr > |z|” below 0.05 or a value of “P > |t|” below 0.05 suggests potential publication bias. Sensitivity analyses were also carried out to determine the effect of individual study on the summary meta-analysis estimate. A *p* value below 0.05 was considered a significant difference in this study.

## 3. Results

### 3.1. Study Selection and Characteristics

Based on the publication searching strategy, we obtained a total of 148 results for preliminary screening, and 11 studies from 10 papers were finally included in this meta-analysis (Figure 1) [7,8,9,10,11,12,13,14,15,16]. Since the paper from Takeshita in 2021 involved two completely independent series of patients and separately reported each serial data, these two studies were separately analyzed in this present study [8]. Among all the 1254 participants, 671 patients were allocated into a DNTP group, 456 patients from 8 studies were allocated into a conventional palpation group, and 207 patients from 3 studies were allocated into a traditional ultrasound method, including SA-OOP and LA-IP (Table 1). Four studies involved patients aged below three years, and the other seven studies involved adult patients. Intravenous catheter of 24G was applied in all the four studies, including small children and infants. In studies of adult patients, 20G were mostly used, and only one study used 22G catheter. The mean catheterized artery diameter varied from 2.3 mm to 2.9 mm in adult patients and 0.85 mm to 1.1 mm in small children and infants. No patient with shock or hypertension was involved in these studies. The methodological quality of included studies was assessed according to the appropriate evaluation criteria by Review Manager 5.4.1, and it was indicated that all included 11 studies were low risk (Figure 2).

### 3.2. Meta-Analysis of DNTP vs. Conventional Palpation or Ultrasound Approach in Efficiency Outcomes

First-attempt success, overall success, and total cannulation time were reported in all included 11 studies while the number of punctures was only reported in 7 of the 11 studies. As shown in Table 2 and Figure 3, compared with conventional palpation, DNTP was associated with significantly increased first-attempt success rate (pooled RR = 1.792, 95% CI: 1.456–2.206, *p* < 0.001) and overall success rate (pooled RR = 1.368, 95% CI: 1.142–1.639, *p* = 0.001). Subgroup analysis by patients’ age resulted in decreased heterogeneity in each group, and it was indicated that the DNTP approach gained even more advantage in small children and infants (first-attempt success rate: 88/120 vs. 30/120, pooled RR = 2.738; overall success rate: 110/120 vs. 63/120, pooled RR = 1.703) than in adults (first-attempt success rate: 293/340 vs. 188/336, pooled RR = 1.514; overall success rate: 325/340 vs. 272/336, pooled RR = 1.168).

It was also shown that the DNTP approach was associated with decreased cannulation time than palpation (65.09 s vs. 142.70 s, pooled SMD = −1.758, 95% CI: −2.766 to −0.750, *p* = 0.001, Table 3 and Figure 3). Further subgroup analysis also indicated that the main advantage in cannulation time of DNTP vs. conventional palpation method appeared in small children and infant patients (63.60 s vs. 342.47 s, pooled SMD = −2.849, 95% CI: −4.503 to −1.194, *p* = 0.001). Number of punctures was mainly reported in studies of adult patients, and it was shown that DNTP was also associated with fewer punctures than conventional palpation (1.16 vs. 1.80, pooled SMD = −0.916, 95% CI: −1.245 to −0.587, *p* < 0.001). Restricted to the smaller number of included studies in this outcome, no subgroup analysis was performed in puncture number.

All the three studies evaluating DNTP, and traditional ultrasound-guided artery cannulation involved adult patients and provided the data of first-attempt success, overall success, cannulation time, and the number of punctures. It was shown that there was no significant difference in first-attempt success (pooled RR = 1.200, 95% CI: 0.980–1.470, *p* = 0.077), overall success (pooled RR = 1.030, 95% CI: 0.974–1.088, *p* = 0.299), cannulation time (pooled SMD = −0.263, 95% CI: −2.306 to 1.779, *p* = 0.800), and the number of punctures (pooled SMD = −0.496, 95% CI: −1.039 to 0.047, *p* = 0.073) between DNTP and a traditional ultrasound approach (Table 3 and Figure 4).

### 3.3. Meta-Analysis of DNTP vs. Conventional Palpation or Ultrasound Approach in Complications

Complications in arterial cannulation were relatively low and not comprehensively recorded and reported in most studies. Hematoma, thrombosis, posterior wall puncture, and vasospasm were mostly reported. As shown in Table 4 and Table 5 and Figure 5, the DNTP approach was associated with less occurrence of hematoma than conventional palpation (pooled RR = 0.265, 95% CI: 0.137–0.514, *p* < 0.001) or traditional ultrasound approach (pooled RR = 0.348, 95% CI: 0.206–0.588, *p* < 0.001). No significant difference in thrombosis was detected between DNTP and palpation (pooled RR = 0.607, 95% CI: 0.079–4.998, *p* = 0.661) or traditional ultrasound approach (pooled RR = 0.232, 95% CI: 0.040–1.357, *p* = 0.105). Additionally, it was indicated that DNPT was associated with fewer arterial posterior wall punctures (pooled RR = 0.495, 95% CI: 0.330–0.744, *p* = 0.001) and vasospasm (pooled RR = 0.267, 95% CI: 0.102–0.697, *p* = 0.007) than traditional ultrasound approach.

## 4. Discussion

This presents study reveals that the DNTP method was superior to conventional palpation in arterial catheterization with substantially increased success rate and reduced complications. DNTP was associated with higher first-attempt success, overall success rate, and fewer number of punctures, cannulation time, and occurrence of hematoma than palpation. However, DNTP did not result in statistically significant decreased cannulation time in adults than palpation method. When compared with traditional ultrasound method, DNTP obtains similar first-attempt success, overall success rate, cannulation time, and number of puncture but much less hematoma, posterior wall puncture, and vasospasm occurrences. Further analysis indicates that DNTP was more preferred in small children and infants.

Cannulation of the artery is frequently used in patients undergoing surgery and ICU patients and could be technically challenging in specific conditions [17]. Previous studies and meta-analysis have indicated benefit from ultrasound-guided artery catheterization compared with conventional palpation method [18]. Traditionally, LA-IP and SA-OOP approaches are mostly applied in clinical practice and assessed by clinical trials. SA-OOP could offer better image of the targeted artery and surrounding tissue while the angle and routine of the puncture needle were not well displayed. LA-IP could show the whole route of the puncture needle and artery orientation, but the surrounding tissues were not well detected [19]. Multiple tries have been taken to overcome their deficiency, and DNTP was a modified method based on the SA-OOP by moving ultrasonic probe proximally along the artery during the cannulation procedure. Thus, the needle tip could be dynamically visualized on the screen during the entire puncture route [20]. For traditional ultrasound method, it was not recommended as first-line use in routine arterial cannulation but a useful rescue technique in difficult arterial cannulation in previous guidelines [21]. Since even fewer studies in the DNTP approach than traditional have been published, no consistent viewpoint has been reached in DNTP as the first line use in arterial cannulation, and we performed this meta-analysis to evaluate the use of DNTP in arterial cannulation.

Clemmesen and his colleagues firstly compared the DNTP and LA-IP method-guided peripheral venous access in phantoms by a randomized study [22]. Hansen and his colleagues then evaluated the value of the DNTP approach in ultrasound-guided radial artery catheterization by a randomized crossover study in undergoing cardiac surgery patients, and it was shown that first-attempt success rate was remarkably increased in the DNTP group than in the conventional palpation method [14]. A previous meta-analysis indicated that DNTP was associated with increased efficiency of first-attempt success except in infant patients [6]. However, this meta-analysis intermixed vein and artery cannulation, and a recent study including infants and small children by Takeshita reached the opposite conclusion [8]. Our present study reveals that DNTP was significantly superior to palpation in artery cannulation with increased success rate and decreased cannulation time. Subgroup analysis indicated more advantage of the DNTP approach in small children and infants with a nearly threefold first-attempt success rate and double overall success rate. This was quite different from previous meta-analysis by Shi [6]. It was shown that the mean artery diameter was around 1 mm (varied from 0.85 to 1.05 mm) in small children and infants while the mean artery diameter in adults was over 2 mm. Taking this into consideration, the ratio of catheter diameter to artery diameter was much higher in small children and infants than in adults even though smaller catheters were applied to small children and infants (24G vs. 20G & 22G). Thus, accurate and dynamic location of the needle and target artery during artery cannulation was more difficult and critical for small children and infants. This could partially explain the result that small children and infants obtained more benefit from the DNTP technique than palpation. These results should be more reasonable and reliable than previous meta-analysis. When compared with the conventional ultrasound approach, no advantages of first-attempt, overall success rate, cannulation time, and number of punctures were acquired by the DNTP approach. These results revealed that DNTP was as effective as the conventional ultrasound method but much better than palpation from the perspective of artery cannulation success rate, especially in small children and infants.

Even though complication incidence of artery cannulation is relatively low, extra attention should be paid in certain scenarios [23]. For critical ill patients receiving heparinization because of intra-aortic balloon pump (IABP) or veno-arterial extracorporeal membrane oxygenation (VA-ECMO) and patients receiving other intensive anticoagulant or thrombolytic therapy, hematoma and posterior wall puncture should be avoided as much as possible [24,25]. Much more than hematoma itself, this could be associated with lower-intensity anticoagulant therapy and increased adverse thrombotic events [26]. Based on our results, the DNTP approach was proven to be associated with decreased incidence of hematoma when compared with either palpation or other traditional ultrasound methods. It was also pointed out that DNTP could also decrease the posterior wall puncture than traditional ultrasound method. These results indicated that DNTP could offer a better choice than conventional palpation or ultrasound method to reduce complications, and DNTP should be preferred in certain patients.

Except for DNTP, other modified methods based on SA-OOP and LA-IP were also practiced in ultrasound-guided arterial cannulation. The modified long-axis in-plane (MLAX-IP) technique included radial artery marked on the skin by repeated short-axis detection and cannulation guided by the long-axis in-plane method. It was shown that MLAX-IP was associated with increased first-attempt cannulation success rate in radial artery catheterization than DNTP [27]. The combined short-axis and long-axis (CSLA) approach also included short-axis detection and guided puncture following a venous entrance by LA-IP. It was indicated that CSLA was associated with decreased posterior wall puncture in central venous catheterization [28]. However, its value in arterial cannulation is still unclear. In total, more studies are still needed to evaluate the value of DNTP and other new methods in artery cannulation.

This meta-analysis also has several limitations. First, although we included more studies than previous meta-analysis, the number is still small and partial outcomes were not reported in some studies. It is difficult to analyze these outcomes by integrating all included studies. Second, since most studies involved patients undergoing elective surgery, the value of DNTP ultrasound-guided artery cannulation in critically ill patients and patients undergoing VA-ECMO or IABP still needs more investigation.

Despite these limitations, there are still some advantages to carrying out this meta-analysis. First, this meta-analysis only included prospective multicenter RCTs and separately analyzed the association between DNTP and different outcomes. This makes the analysis more accurate and reliable. Second, by including the most recent study, this meta-analysis involved the most participants and RCTs about DNTP and artery cannulation. This makes the result more robust. Third, we evaluate the effect of DNTP in distinct subgroups and obtain decreased heterogeneity in each subgroup. This helps to offer innovative results in small children and makes the results more reliable.

## 5. Conclusions

In conclusion, this meta-analysis indicates that the DNTP approach was a better choice in artery cannulation than conventional palpation and ultrasound methods, especially in small children and infants. Well-designed RCTs are in need to verify this conclusion.

## Figures and Tables

**Figure 1 jcm-11-06539-f001:**
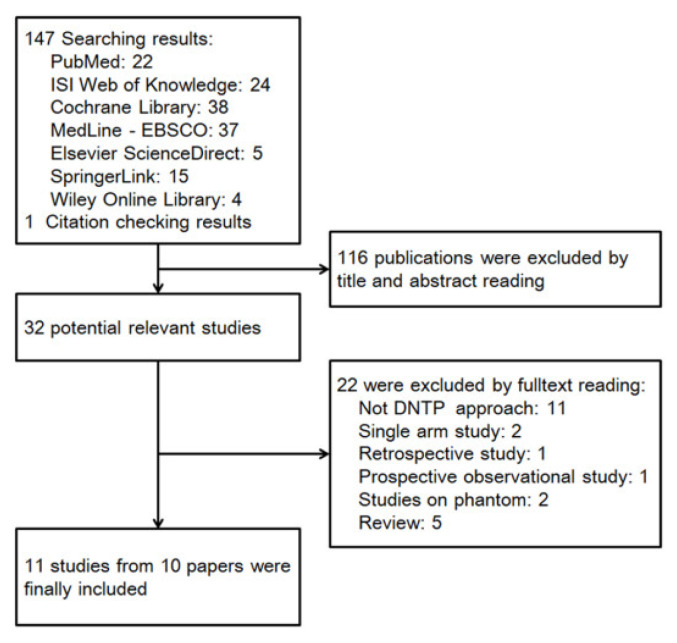
Flow diagram of selected studies.

**Figure 2 jcm-11-06539-f002:**
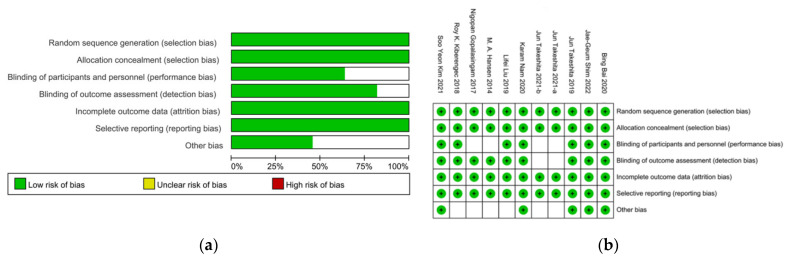
Risk of bias assessment in randomized controlled trial studies by RevMan. (**a**) Risk of bias graph. (**b**) Risk of bias summary [7,8,9,10,11,12,13,14,15,16].

**Figure 3 jcm-11-06539-f003:**
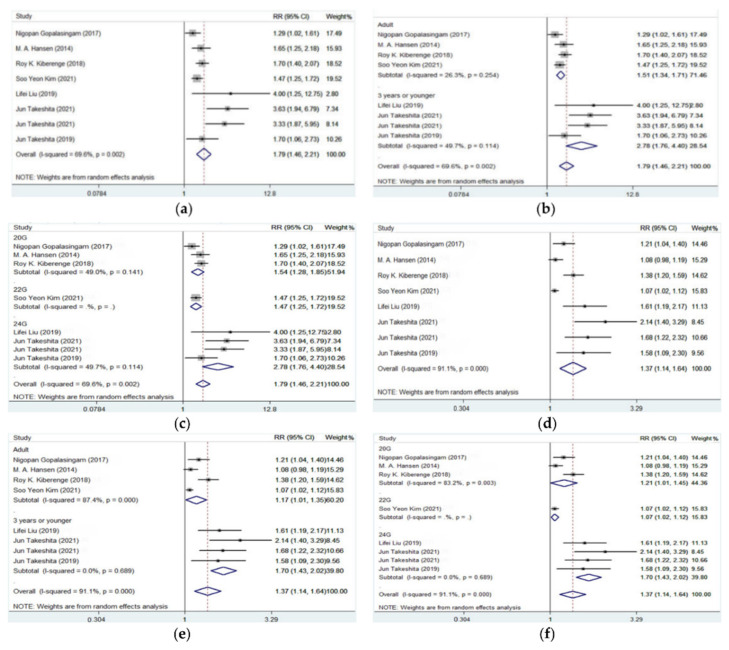
Forest plots for the efficiency outcomes of artery catheterization guided by DNTP vs. palpation approach [7,8,11,12,13,14,15]. (**a**) First-attempt success of DNTP vs. palpation; (**b**) subgroup analysis by patient age in first-attempt success of DNTP vs. palpation; (**c**) subgroup analysis by catheter size in first-attempt success of DNTP vs. palpation; (**d**) overall success of DNTP vs. palpation; (**e**) subgroup analysis by patient age in overall success of DNTP vs. palpation; (**f**) subgroup analysis by catheter size in overall attempt success of DNTP vs. palpation; (**g**) cannulation time of DNTP vs. palpation; (**h**) subgroup analysis by patient age in cannulation time of DNTP vs. palpation; (**i**) subgroup analysis by catheter size in cannulation time of DNTP vs. palpation; (**j**) puncture number of DNTP vs. palpation; (**k**) subgroup analysis by patient age in puncture number of DNTP vs. palpation.

**Figure 4 jcm-11-06539-f004:**
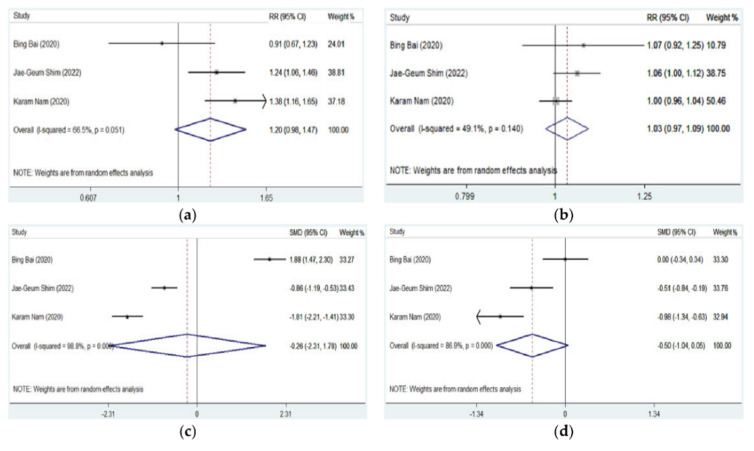
Forest plots for the efficiency outcomes of artery catheterization guided by DNTP vs. traditional ultrasound approach [9,10,16]. (**a**) First-attempt success of DNTP vs. traditional ultrasound; (**b**) overall success of DNTP vs. traditional ultrasound; (**c**) cannulation time of DNTP vs. traditional ultrasound; (**d**) puncture number of DNTP vs. traditional ultrasound.

**Figure 5 jcm-11-06539-f005:**
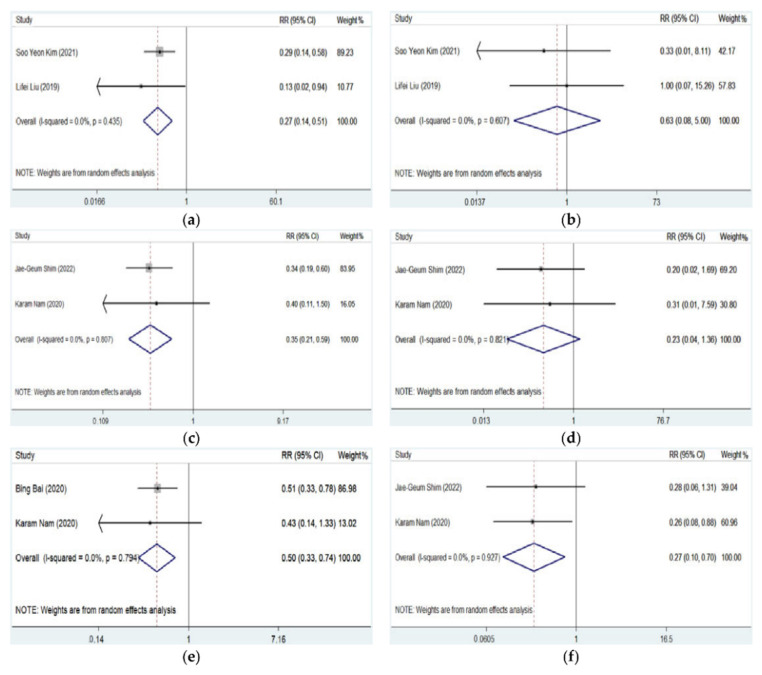
Forest plots for complications of artery catheterization guided by DNTP vs. palpation or traditional ultrasound approach [9,10,11,12,16]. (**a**) Occurrence of hematoma of DNTP vs. palpation; (**b**) occurrence of thrombosis of DNTP vs. palpation; (**c**) occurrence of hematoma of DNTP vs. traditional ultrasound; (**d**) occurrence of thrombosis of DNTP vs. traditional ultrasound; (**e**) occurrence of posterior wall punctures of DNTP vs. traditional ultrasound; (**f**) occurrence of vasospasm of DNTP vs. traditional ultrasound.

**Table 1 jcm-11-06539-t001:** Main characters of included studies.

	Year	Country	Catheter Size	Artery	Groups	Patient Details	Total Number	Reference
Bing Bai	2020	China	20G	Radial artery	DNTP vs. SA-OOP	Adult	131	[16]
Nigopan Gopalasingam	2017	Denmark	20G	Radial artery	DNTP vs. P	Adult	40	[15]
M. A. Hansen	2014	Denmark	20G	Radial artery	DNTP vs. P	Adult	40	[14]
Roy K. Kiberenge	2018	America	20G	Radial artery	DNTP vs. P	Adult	260	[13]
Soo Yeon Kim	2021	Korea	22G	Radial artery	DNTP vs. P	65 years or older	256	[12]
Lifei Liu	2019	China	24G	Radial artery	DNTP vs. P	Neonate	60	[11]
Jae-Geum Shim	2022	Korea	20G	Radial artery	DNTP vs. SA-OOP	70 years or older	151	[9]
Jun Takeshita	2021	Japan	24G	Posterior tibial artery	DNTP vs. P	3 years or younger	70	[8]
Jun Takeshita	2021	Japan	24G	Dorsalis pedis artery	DNTP vs. P	3 years or younger	70	[8]
Jun Takeshita	2019	Japan	24G	Deep artery	DNTP vs. P	3 years or younger	40	[7]
Karam Nam	2020	Korea	20G	Radial artery	DNTP vs. LA-IP	Adult	136	[10]

Note: DNTP: dynamic needle-tip positioning; SA-OOP: short axis out-of-plane; LA-IP: long axis in-plane.

**Table 2 jcm-11-06539-t002:** Meta-analysis and subgroup analysis in success rate of DNTP vs. conventional palpation or ultrasound approach.

Outcome	Study Included (*n*)	Case Number (*n*)		Heterogeneity		Pooled RR	95% CI	*p*-Value
		DNTP	Control	I-Squared (%)	*p*			
**DNTP vs. palpation**								
**First-attempt success**	8	460	456	69.6	0.002	1.792	1.456–2.206	<0.001
Grouped by patient age								
Adult	4	340	336	26.3	0.254	1.514	1.341–1.708	<0.001
≤3 years	4	120	120	49.7	0.114	2.783	1.762–4.396	<0.001
Grouped by catheter size								
20G	3	212	208	49	0.141	1.537	1.276–1.851	<0.001
22G	1	128	128	-	-	1.467	1.248–1.724	<0.001
24G	4	120	120	49.7	0.114	1.792	1.456–2.206	<0.001
**Overall success**	8	460	456	91.1	<0.001	1.368	1.142–1.639	0.001
Grouped by patient age								
Adult		340	336	87.4	<0.001	1.168	1.013–1.347	0.033
≤3 years		120	120	0	0.689	1.703	1.433–2.024	<0.001
Grouped by catheter size								
20G	3	212	208	83.2	0.003	1.211	1.011–1.452	0.038
22G	1	128	128	-	-	1.067	1.015–1.122	0.011
24G	4	120	120	0	0.689	1.703	1.433–2.024	<0.001
**DNTP vs. traditional ultrasound**								
**First-attempt success**	3	211	207	66.5	0.051	1.200	0.980–1.470	0.077
**Overall success**	3	211	207	49.1	0.140	1.030	0.974–1.088	0.299

Note: DNTP: dynamic needle-tip positioning; RR: relative risk.

**Table 3 jcm-11-06539-t003:** Main results of meta-analysis in arterial catheterization practice of DNTP vs. palpation or traditional ultrasound approach.

Outcome	Study Included (*n*)	Case Number (*n*)		Mean		I-Squared (%)	Pooled SMD	95% CI	*p*-Value
		DNTP	Control	DNTP	Control				
**DNTP vs. palpation**									
**Cannulation time**	8	460	456	65.09s	142.70s	97.5	−1.758	−2.766 to −0.750	0.001
Grouped by patient age									
Adult	4	340	336	65.61s	70.51s	98.1	−0.756	−2.006 to 0.494	0.236
≤3 years	4	120	120	63.60s	342.41s	95.4	−2.849	−4.503 to −1.194	0.001
Grouped by catheter size									
20G	3	212	208	79.87s	81.09s	95.2	−0.344	−1.362 to 0.674	0.508
22G	1	128	128	42s	53s	-	−1.953	−2.251 to −1.655	<0.001
24G	4	120	120	63.60s	342.47s	95.4	−1.758	−2.766 to −0.750	0.001
**Number of puncture**	5	360	356	1.16	1.80	73.2	−0.916	−1.245 to −0.587	<0.001
**DNTP vs. traditional ultrasound**									
**Cannulation time**	3	211	207	62.52	64.30	98.8	−0.263	−2.306 to 1.779	0.800
**Number of puncture**	3	211	207	1.20	1.44	86.9	−0.496	−1.039 to 0.047	0.073

Note: DNTP: dynamic needle-tip positioning; SMD: standardized mean difference.

**Table 4 jcm-11-06539-t004:** Meta-analysis and subgroup analysis in complications of DNTP vs. palpation or traditional ultrasound method.

Outcome	Study Included (*n*)	Case Number (*n*)		Heterogeneity		Pooled RR	95% CI	*p*-Value
		DNTP	Control	I-Squared (%)	*p*			
**DNTP vs. palpation**								
Hematoma	2	158	158	0	0.435	0.265	0.137–0.514	<0.001
Thrombosis	2	158	158	0	0.607	0.629	0.079–4.998	0.661
**DNTP vs. traditional ultrasound**								
Hematoma	2	146	141	0	0.807	0.348	0.206–0.588	<0.001
Thrombosis	2	146	141	0	0.821	0.232	0.040–1.357	0.105
Posterior wall puncture	2	135	132	0	0.794	0.495	0.330–0.744	0.001
Vasospasm	2	146	141	0	0.927	0.267	0.102–0.697	0.007

Note: DNTP: dynamic needle-tip positioning; RR: relative risk.

**Table 5 jcm-11-06539-t005:** Detailed information of complications reported in each study.

Study	Year	Artery	Groups	Hematoma	Thrombosis	Posterior Wall Puncture	Vasospasm
Bing Bai [16]	2020	Radial artery	DNTP vs. SA-OOP			19/67 vs. 37/67	
Nigopan Gopalasingam [15]	2017	Radial artery	DNTP vs. Palpation				
M. A. Hansen [14]	2014	Radial artery	DNTP vs. Palpation				
Roy K. Kiberenge [13]	2018	Radial artery	DNTP vs. Palpation				
Soo Yeon Kim [12]	2021	Radial artery	DNTP vs. Palpation	9/128 vs. 31/128	0/128 vs. 1/128		0/128 vs. 0/128
Lifei Liu [11]	2019	Radial artery	DNTP vs. Palpation	1/30 vs. 8/30	0/30 vs. 0/30		
Jae-Geum Shim [9]	2022	Radial artery	DNTP vs. SA-OOP	12/76 vs. 35/75	1/76 vs. 5/77		2/76 vs. 7/75
Jun Takeshita [8]	2021	Posterior tibial artery	DNTP vs. Palpation				
Jun Takeshita [8]	2021	Dorsalis pedis artery	DNTP vs. Palpation				
Jun Takeshita [7]	2019	Deep artery	DNTP vs. Palpation				
Karam Nam [10]	2020	Radial artery	DNTP vs. LA-IP	3/70 vs. 7/66	0/70 vs. 1/66	4/68 vs. 9/66	3/70 vs. 11/66

Note: DNTP: dynamic needle-tip positioning; SA-OOP: short axis out-of-plane; LA-IP: long axis in-plane.

## Data Availability

Reported results can be found from original paper as stated in reference.

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
