# Peer review of "Ultrasound-Guided Dynamic Needle-Tip Positioning Method Is Superior to Conventional Palpation and Ultrasound Method in Arterial Catheterization"

_jcm, 2022, doi:10.3390/jcm11216539_

Round 1

Reviewer 1 Report

The authors presented a meta-analysis in which they evaluated studies comparing the "Dynamic needle-tip positioning" management used in the management of arterial cannulation with other classical methods. Arterial cannulation is an intervention that is used intensively especially in intensive care units and is difficult to apply especially in pediatric patients.The authors also came to this conclusion in their study. I think the presented article is important and original for readers.

Author Response

Dear reviewer,

Thank you and grateful for your valuable comments. We prepared this paper out of troubles in clinical practice of artery cannulation in some patients. We hope this could help to offer some assist in this field.

Thank you again for your kindly comments.

Reviewer 2 Report

1. This is a very informative meta-analysis

2. On page two of the manuscript the second word in second line is uninterpretable: “farraginious”

                Please correct.

3.  Table 1:

                It would be helpful to the reader if you supplied the reference citation number for each of these studies

4. Page 10, Discussion, first paragraph.

                a. I suggest delete “and ultrasound guided” from the first sentence.

                b. This is because you did not find that DNTP improved first attempt or overall success rate or cannulation time compared with traditional ultrasound guided techniques.

                c. You did clarify that DNTP compared with conventional ultrasound guidance was associated with less hematomas, posterior wall puncture, and vasospasm.

                d. You might mention that compared with palpation, DNTP was not associated with a statistically significant decrease in cannulation time in adults.

5. In your discussion you may wish to refer to and comment on the guidelines for performing ultrasound guided vascular cannulation: recommendations of the American Society of Echocardiography and the Society of Cardiovascular Anesthesiologists, by Troianos et al published in 2011. [Troianos CA, et al.  Guidelines for performing ultrasound guided vascular cannulation: recommendations of the American Society of Echocardiography and the Society of Cardiovascular Anesthesiologists. Am Soc Echocardiogr. 2011 Dec; 24(12):1291-318.]

            Based upon the data available at that time, these authors did not recommend real-time ultrasound use for arterial cannulation. However, they mentioned that for radial artery   cannulation, there was category A, level 1 support for use of ultrasound to improve first pass success, referring to the earlier smaller meta-analysis by Shiloh, et al (your citation number 3). They also discussed its usefulness as a rescue technique and to identify the location and patency of suitable arteries, and to identify tortuosity, atheromatous plaques during difficult catheter insertion. (See their recommendation 11.3)

Author Response

Dear reviewer,

Thank you and deeply appreciate your valuable comments. We have modified the manuscript according to your constructive suggestions and answered each question point to point. 
Please see the attachment.

Thank you again for your kindly comments.

Reviewer 3 Report

The purpose of this review is to compare and highlight the superiority of Dynamic needle-tip positioning method regarding complications and efficacy.

In my point of view this is an interesting review, using nowadays knowledge and studies, though there are some comments I would like to point.

1.       It is quite clear from the authors that DNTP should be preferred in specific group of patients, mainly infants and children, and in patients under ECMO or IABP. However, it is not yet clear the usefulness of the current method in the critical ill patients, and this review does not provide sufficient information on this population.

2.       Moreover, the site of the arterial puncture is not specified through the review. As, for example puncture of the femoral artery can be related with major complications, it is important to be noticed, and defined the site of puncture. A new table for the different puncture sites and the percentage of complications in each site could give more extended and accurate information to the reader.

Author Response

Dear reviewer,

Thank you and deeply appreciate your constructive  comments. We have modified the manuscript according to your valuable suggestions and answered each question point to point. 
Please see the attachment.

Thank you again for all your precious suggestions which help to make this paper better.
